# Macrophages as a Potential Immunotherapeutic Target in Solid Cancers

**DOI:** 10.3390/vaccines11010055

**Published:** 2022-12-26

**Authors:** Alok K. Mishra, Shahid Banday, Ravi Bharadwaj, Amjad Ali, Romana Rashid, Ankur Kulshreshtha, Sunil K. Malonia

**Affiliations:** 1Department of Molecular, Cell and Cancer Biology, UMass Chan Medical School, Worcester, MA 01605, USA; 2Department of Medicine, UMass Chan Medical School, Worcester, MA 01605, USA

**Keywords:** TAMs, immunotherapy, inflammation, prognosis, metastasis, phagocytosis, drug resistance, CD47, CSF1R, cancer stem cells, CAR macrophages, clinical trials

## Abstract

The revolution in cancer immunotherapy over the last few decades has resulted in a paradigm shift in the clinical care of cancer. Most of the cancer immunotherapeutic regimens approved so far have relied on modulating the adaptive immune system. In recent years, strategies and approaches targeting the components of innate immunity have become widely recognized for their efficacy in targeting solid cancers. Macrophages are effector cells of the innate immune system, which can play a crucial role in the generation of anti-tumor immunity through their ability to phagocytose cancer cells and present tumor antigens to the cells of adaptive immunity. However, the macrophages that are recruited to the tumor microenvironment predominantly play pro-tumorigenic roles. Several strategies targeting pro-tumorigenic functions and harnessing the anti-tumorigenic properties of macrophages have shown promising results in preclinical studies, and a few of them have also advanced to clinical trials. In this review, we present a comprehensive overview of the pathobiology of TAMs and their role in the progression of solid malignancies. We discuss various mechanisms through which TAMs promote tumor progression, such as inflammation, genomic instability, tumor growth, cancer stem cell formation, angiogenesis, EMT and metastasis, tissue remodeling, and immunosuppression, etc. In addition, we also discuss potential therapeutic strategies for targeting TAMs and explore how macrophages can be used as a tool for next-generation immunotherapy for the treatment of solid malignancies.

## 1. Introduction

In 1882, Russian zoologist Elie Metchnikoff, who shared the 1908 Nobel Prize in immunology with Paul Ehrlich, identified the first ever cell with prolongations that had the capacity to engulf foreign particles. He found that these cells had the capacity to respond to infections and foreign particles in a way that is analogous to inflammation in higher species. Metchnikoff named them phagocytes [Metchnikoff’s citation; *‘since these cells aside from their capacity to stretch out prolongations also are capable of consuming foreign bodies, we will subsume them under the joint name of fagocytes [sic].”*]. Later, these cells were termed as macrophages [1,2]. As a part of the innate immune system, macrophages constitute the first line of defense against the invading pathogens by recognizing pathogen-associated molecular patterns (PAMPs) through pattern recognition receptors (PRRs) [3]. Macrophages engulf the pathogen in the cytoplasm, process the antigen through proteolytic cleavage, load the processed antigen on MHC class II, and display them on the cell surface. This process is called antigen presentation. Macrophages act as professional antigen-presenting cells (APCs) and convey the antigenic signal to helper T-cells, thereby activating the adaptive immune system. The innate immune system can also destroy altered or malignant cells in the same manner as it does with foreign pathogens.

The increasing evidence suggests that macrophages play a crucial role in both subversion as well as the progression of cancer [4]. They have the potential to prevent tumor initiation and progression by using one or more mechanisms, such as antibody-dependent cellular cytotoxicity (ADCC), antibody-dependent cellular phagocytosis (ADCP), phagocytosis by recognizing prophagocytic signals on neoplastic cells, inducing vascular damage and tumor necrosis, and activation of tumor defense by adaptive lymphoid cells through antigen presentation [4,5,6]. Contrarily, the macrophages that infiltrate tumor microenvironments (TMEs) are differently activated by the suppressive cytokines and chemokines produced by tumor cells and programmed to support tumor growth and metastasis. These macrophages are called tumor-associated macrophages (TAMs). TAMs promote tumor progression in several ways, such as supporting the survival and proliferation of cancer cells, promoting neovascularization, establishing an immunosuppressive TME, and increasing genetic instability, tumor fibrosis, invasion, and metastasis [6,7,8,9,10,11,12]. Moreover, macrophages are also responsible for resistance to chemotherapy and immunotherapy given for cancer treatment.

Therapeutic targeting of TAMs as a cancer treatment strategy has shown promising results in preclinical studies as well in clinical trials [8,11]. Enhancing phagocytosis by checkpoint blockade or opsonization, preventing M2 polarization, depleting tumor-associated macrophages, re-educating or inhibiting TAM recruitment, cytokine delivery, targeting anti-inflammatory signaling and TAM metabolism, and using macrophages as cell therapy, which has also advanced to clinical testing, are the available therapeutic implications of macrophages [11,13,14,15]. In this review, we provide a comprehensive overview on the pathobiology of TAMs in cancers. We discuss various pro-tumorigenic functions of TAMs and strategies that are currently being evaluated for targeting TAMs to enhance tumor elimination. We also explore how macrophages can be used as a tool for next-generation cell-based immunotherapies for solid tumors.

## 2. Origin and Phenotypic Plasticity of TAMs

TAMs have long been believed to be originated from bone marrow-derived monocytic precursors which migrate into the TME by chemotaxis [16,17]. Recent studies indicate that tissue-specific embryonically derived macrophages also infiltrate tumor tissues and constitute a significant source of TAMs [8,16,17]. Macrophages are one of the most plastic immune cells, capable of acquiring distinct functional phenotypes depending on the microenvironment signals (Figure 1A). They can reversibly transform into anti-tumorigenic M1-like or pro-tumorigenic M2-like phenotypes in response to specific stimuli in the TME. The Th1 cytokines (IFNγ, TNF-α,) and bacterial-derived LPS promote the differentiation of monocytes into M1 phenotype. Whereas M2 differentiation occurs in response to Th2 cytokines (IL-4, IL-10, TGF-β1) and PGE2 produced in the TME [8,10,16]. The M1 and M2 division of TAMs is based on the surface markers expressed by these two types of macrophages. HLA-DR, CD80, CD86, CD197, TLR-2,4, and iNOS are dominant surface markers of M1 macrophages, whereas the M2 macrophage express CD163, CD209, CD206, FIZZ1, Ym1/2, and CCL2 are on their surface. The M1 macrophages retain their intrinsic ability to phagocytose and produce anti-tumor inflammatory responses [10]. The efficient tumor antigen-presenting ability of M1 macrophages promotes recruitment and stimulation of other leukocytes to exert their cytotoxic functions against cancer cells. For example, the immunostimulatory cytokines (IL-6, IL-12, TNF) released by M1 macrophages can enhance the activity of CD8 + T-cells and NK cells (Figure 1A). On the other hand, M2-type macrophages are endowed with a variety of tumor-promoting capabilities including immunosuppression, angiogenesis, neovascularization, as well as the activation of stromal cells [8,10,16,17,18]. However, it is worth mentioning that although TAMs are often considered identical to M2-type macrophages, a transcriptomic analysis of TAMs isolated from a murine fibrosarcoma revealed an unexpected transcriptomic profile of TAMs that differed from both M1 and M2 macrophages. In addition, an individual macrophage (TAM) can co-express markers of M1 and M2 phenotypes [19]. TAM diversity in tumors, therefore, is much more complex than the overly simplistic classification of TAMs into M1- and M2-type [20].

## 3. Prognostic Significance of TAMs in Solid Tumors

TMEs are enriched with immune cells during tumor development. Tumor immune infiltrates, however, are dominated by macrophages. In fact, up to 50% of the tumor mass is composed of TAMs [21]. Since macrophages have both proinflammatory (M1-type) and anti-inflammatory (M2-type) functions, analyzing their role in cancer progression is overly complex [7,21,22]. In terms of expression profile and phenotype, TAMs differ from peripheral or tissue-resident macrophages. Unlike classical macrophages, which typically differentiate into M1-type, the major proportion of TAMs is differentiated into M2-type due to suppressive cytokines released by tumor cells in the TME or by engaging receptors of tumor-infiltrating macrophages with immune checkpoint proteins displayed on tumor cells. The density of a TAM has been linked to a poor prognosis in various solid malignancies, including breast [23], bladder, prostate, head and neck, glioma, melanoma, thyroid, lung, hepatocellular cancers, and non-Hodgkin lymphoma [12]. According to meta-analyses of studies on the association of TAM with cancer prognosis, macrophage density is found to be significantly correlated with poor patient outcomes [23,24,25]. Recent studies in multiple cancer models further supported these analyses [26,27]. Interestingly, in some cancers, such as gastric cancer, colorectal cancer, and bladder cancer, TAMs density has also been found to be positively correlated with favorable disease outcomes [28,29] (Figure 1b). The reason behind this differential behavior of TAMs in a cancer type-specific manner remains to be defined. Moreover, it is not always the case that tumor-promoting TAMs have the M2-type phenotype; therefore, the sub-typing of TAMs beyond M1/M2 dichotomy is also needed. Though the existence of subtypes other than M1/M2 subsets, such as regulatory macrophages (Mreg), wound-healing macrophages (IL-4 activated macrophages, (_M (IL-4)_) and hybrid macrophages (IL-4-primed macrophages treated with lipopolysaccharide (LPS) and immune complexes), have been suggested by many investigators [30]. However, extensive in vivo studies are needed to identify the TAMs subtypes in tumors and their role in tumor progression. In addition, histopathological distributions of TAM within cancer tissue are also relevant to the outcome of cancer [10,31].

## 4. Pro-Tumorigenic Roles of TAMs

Until recently, TAMs were presumed to be just passive bystanders which have lost their ability to regulate cancers owing to an immunosuppressive microenvironment. However, several recent evidence point out that TAMs may be actively engaged in fueling tumor growth. Understanding the mechanism(s) that TAMs employ to promote tumorigenesis becomes essentially important. There are multiple mechanisms known by which TAM may promote tumorigenesis. The sections given below highlight the pro-tumorigenic role played by TAMs in cancers (Figure 2).

### 4.1. TAM-Mediated Inflammation and Genomic Instability

The association between inflammation and cancer dates to 1863 when Rudolf Virchow hypothesized that cancer originated at sites of chronic inflammation. His hypothesis was based in part on the fact that some classes of irritants, along with tissue damage and inflammation, promote cell proliferation [32]. Macrophages play a crucial role in the progression as well as the resolution of inflammation. Their role in cancer-associated inflammation is also widely acknowledged. The process of tumor growth mimics the phenomenon of wound heating where a rapid proliferation of cells is required for the healing of the wounds which is catalyzed by inflammatory mediators. In the process of wound healing, monocytes migrate to the site of injury and mature into macrophages and become the source of growth factors and cytokines including TGF-β1, PDGF, bFGF, TGF-α, IGF-I/II, TNF-α, and IL-1. These factors modulate tissue repair after inflammation. A large body of evidence highlights that cancer-related inflammation is regarded as a key aspect of the development of tumors. The presence of chronic and unresolved inflammation has been linked to the risk of malignancies and tumor progression in many types of cancer [32]. The best example of chronic inflammation associated with malignant diseases can be the frequent occurrence of colon carcinogenesis in individuals with chronic ulcerative colitis and Crohn’s disease [33]. In addition, the association of Hepatitis C infection with liver cancer and *Helicobacter pylori* infection with gastric cancer has been widely known [34,35,36]. An inflammation-induced genomic insult is thought to be the cause of *Helicobacter pylori* infection-mediated gastric cancer [35,36]. It has been found that macrophage migration inhibitory factor (MIF) produced by macrophages directly enhances inflammation-induced DNA damage [37,38]. In addition, nitric oxide (NO) and reactive oxygen intermediates (ROI) released by TAMs can also damage DNA and cause genetic instability that can lead to tumor formation [39].

### 4.2. Tumor Growth Promotion

The role of TAMs in promoting tumor initiation and development has become increasingly evident over the years. TAMs not only assist cancer development, but also initiate tumor growth through the release of signal molecules and extracellular vehicles (EVs). Macrophages secrete various growth factors such as TGF-β, VEGF, PDGF, cytokines such as M-CSF and interleukins, chemokines such as IL-6, IL-10, CCL2, and CXCL, and enzymes such as MMPs which promote tumor growth [9]. TGF-β1 secreted by TAMs promotes the proliferation and invasion of colorectal cancer by modulating the MIR-34a/VEGF axis [40]. By activating STAT3, TAM-derived IL-6 promotes hepatocellular carcinoma (HCC) development [41]. CCL2 facilitates tumor growth by activating the PI3K/Akt/mTOR in breast cancer [42]. A recent study found that MMP1 derived from TAMs enhanced colon cancer cell proliferation by accelerating cell cycle transitions from G0/G1 to S and G2/M [43].

### 4.3. Formation of Cancer Stem Cells (CSC)

The role of TAMs in CSC expansion has been well-established and is attributed to the TAMs ability to secrete cytokines, and chemokines, growth factors, and exosomes enrich the CSC niche and improve their ability to maintain stem-like properties [44,45]. Through the release of IL-10, M2-type macrophages induce CSC-like properties in non-small cell lung cancer (NSCLC) cells through the activation of JAK1/STAT1/NF-κB/Notch1 signaling [46]. The IL-6 production by TAMs has been found to be associated with the expansion of CSCs in HCC via activation of STAT3 signaling [47]. Moreover, IL-8, epidermal growth factor (EGF), and milk fat globule-EGF factor 8 protein (MFG-E8 protein) derived from TAMs also activate the STAT3 pathway that further increases the stemness of CSCs [48]. IL-8 can also enhance the self-renewal capacity of CSCs in breast cancer patients by stimulating CXCR1 and CXCR2 receptors [48,49]. The existence of stem cells in lung cancer is associated with the production of MUC1 by TAMs [50]. Further, the chemokine CCL2, produced by TAMs, induces an expression of the stem cell regulatory transcription factor, SOX2, OCT3/4, and NANOG via activation of β-Catenin signaling to support the expansion of breast CSCs [51]. The inflammatory mediator Prostaglandin E2 (PGE2), produced by TAMs, promotes the expansion of colon CSCs by activation of NF-κB via EP4-PI3K and EP4-mitogen-activated protein kinase signaling [52]. By expressing corresponding protein molecules on the cell surface, such as CD90, EphA4, and SIRPα, TAMs can also directly bind to CSCs to support CSC stemness [13,53]. It has been shown that extracellular vesicles produced by TAMs also contribute to tumor invasion. Macrophages-derived extracellular vesicles containing MicroRNA-21-5p induce Pa-CSC differentiation by coordinating with KLF3 [54]. The depletion of TAMs has been found to be associated with the reduced frequency of CSCs. By targeting the colony-stimulating factor-1 receptor (CSF-1R) or CXCR2, the inhibition of TAM recruitment resulted in a STAT3-dependent decrease in CSC numbers in a mouse model of pancreatic ductal adenocarcinoma (PDAC) [55].

### 4.4. Epithelial to Mesenchymal Transition (EMT) and Metastasis

In addition to intrinsic growth-promoting changes within neoplastic cells, fertile ground is essential for tumor progression and metastasis. TAMs play a crucial role in the process of inducing EMT, as well as providing metastatic ground. During the EMT process, the interaction between the tumor cell and the TAMs plays a pivotal role. TAMs-derived TGF-β, TNF-α, CCL18, IL-6, IL-8, and IL-10 have been linked to EMT induction and metastasis by regulating various intracellular pathways such as TGF-β-SMAD signaling, MAPK signaling, WNT-β-Catenin pathway, NF-KB signaling, and PI3K-AKT signaling pathway. By inducing theses signaling pathways, TAMs promote the expression of mesenchymal cell markers while inhibiting the expression of epithelial cell markers, resulting in EMT in tumor cells. TAMs-secreted TGF-β binds to its receptors TGF-βR1 and TGF-βR2 and phosphorylates SMAD2 and SMAD3 proteins, which then combines with SMAD4 and forms the trimeric SMAD complex. The subsequent translocation of this trimeric SMAD complex into nucleus regulates the expression of EMT-associated genes through transcriptional mechanisms [56]. Similarly, TNF-α induces EMT by inhibiting the expression of epithelial marker E-cadherin, upregulating the expression of mesenchymal markers, such as vimentin, N-cadherin, and fibronectin, and activating matrix metalloproteinase-9 (MMP-9) by interacting with its receptors TNFR1 and TNFR2, thereby inducing different signaling pathways [57]. Moreover, TGF-β and TNF-α released from macrophages work collaboratively to induce EMT. A synergistic role of TGF-β and TNF-α has been found to induce EMT-mediated breast cancer cell migration and metastasis [58,59]. CCL18 produced by TAMs promotes breast cancer metastasis by downregulating miR98 and miR27b expression via the N-Ras/ERK/PI3K/NF-κB/Lin28b signaling pathway by inducing EMT [60]. IL-8 promotes the migration and EMT of triple-negative breast cancer (TNBC) and ovarian cancer cells via PI3K-Akt signaling and the WNT/β-catenin pathway, respectively [61,62]. The inflammatory cytokine IL-6 secreted by TAMs induces EMT and promotes tumor cell invasion in lung cancer via the COX-2/PGE2/β-catenin signaling pathway [63]. IL-10 is another cytokine that is abundantly produced by TAMs. TAMs release IL-10 upon activation of TLR4 signaling. The TLR4/IL-10 signaling pathway has been found to be involved in the promotion of EMT in pancreatic cancer cells [64]. Moreover, M2-polarized macrophages promote the migration and EMT of HCC cells via the TLR4/STAT3 signaling pathway [65]. In addition, tumor hypoxia induced an HIF-1α/IL-1β signaling loop between cancer cells and TAMs that leads to EMT in HCC [66]. EGF derived from TAMs induces EMT in head and neck squamous cell carcinoma by activating the EGFR/ERK1/2 signaling pathway [15,67]. Moreover, M2-like TAMs secrete CCL20 to activate CCR6 in cancer cells, thereby enhancing the metastasis of primary cutaneous melanoma tumors [68]. In addition, a long noncoding RNA (LncRNA) AFAP1-AS1 in the exosome derived from TAMs induces EMT-associated gene expression and metastasis in esophageal cancer by downregulating miRNA-26a and thereby upregulating its target transcription factor ATF2 [15,69].

### 4.5. Angiogenesis

Angiogenesis or blood vessel formation plays a pivotal role in tumor growth and metastasis. Vasculature promotes cancer cell proliferation by oxygenating and nourishing them. It is crucial for tumors to acquire vascularization to grow beyond 1–2 mm in diameter [70]. TAMs contribute significantly to the formation of new blood vessels in solid tumors [4,71]. TAMs release considerable amounts of VEGF-A, which is a major angiogenesis factor. A positive correlation between high levels of VEGF and TAM density has been found in several cancer types. Immunohistochemical analysis of human invasive ductal breast carcinoma samples found a positive correlation between TAM level and high VEGF expression and micro vessel density (MVD) [72,73]. VEGF secreted by M2-type macrophages and malignant cells in the TME induces endothelial cells to produce factors required for angiogenic growth. VEGF forms a positive feedback loop between angiogenesis and the polarization of macrophages. Tumor-derived VEGF induces macrophage polarization, which then secretes more VEGF to further induce angiogenesis. In a recent study, inhibition of VEGF prevented TAM polarization and attenuated the TAM-mediated secretion of tumor-promoting cytokines [74]. In addition, the presence of VEGF receptors on TAMs also forms an autocrine loop that receives autocrine signals upon VEGF binding with the receptors and enhances the pro-angiogenic properties of TAMs. In an orthotopic ovarian cancer mouse model it has been demonstrated that depletion of TAMs by using clodronate liposomes attenuated tumor growth through the inhibition of angiogenesis [75]. Moreover, TAMs also secrete other proangiogenic growth factors such as EGF, PDGF, FGF2, TGF-β, TNF-α, semaphorin 4D, adrenomedullin, thymidine phosphorylase, IL-6, IL-1*β*, IL-8, CCL2, CXCL8, and CXCL12 which promote neovascularization at several steps including supporting the proliferation of endothelial cells (eCs), as well as inducing sprouting, tube formation, and maturation of new blood vessels [4,11,76,77,78,79,80,81,82]. In addition, there are several enzymes that are often expressed by TAMs including MMP-2, MMP-7, MMP-9, MMP-12, and cyclooxygenase-2 that can have a profound influence on tumor angiogenesis [83].

### 4.6. Treatment Resistance

TAMs in tumors that have undergone chemotherapy can have both positive and negative treatment outcomes. They can occasionally enhance treatment effectiveness, but more frequently cause treatment resistance. Certain drugs stimulate anti-tumor immunity through a mechanism of immunological cell death where TAMs play the central role in immune stimulation [84,85]. For instance, doxorubicin’s therapeutic efficacy is enhanced by TAMs [86]. On the other hand, there are considerable evidence that TAMs induce tumor chemotherapy resistance through the release of inflammatory cytokines and chemokines such as IL-6, CCL18, TNF-α, CCL2, and CXCR4 [18]. IL-6 is a pleiotropic cytokine that induces cancer chemotherapy resistance in various ways. TAMs also contribute to treatment resistance by stimulating CSCs and EMT, as well as metabolic reprogramming to promote angiogenesis [44,87,88]. In addition, TAMs foster resistance to immunotherapy by expressing immune checkpoints, promoting an immunosuppressive TME that limits T-cells infiltration and effector functions [89,90]. TAMs block monoclonal antibody treatment against immune checkpoints by sequestering them by their cell surface Fcγ receptors. According to a recent study, monoclonal antibodies against PD-1 bind to tumor-infiltrating effector T-cells initially, but are absorbed by TAMs during treatment, diminishing the impact of the therapy. Further, a blockade of Fcγ receptors before PD-1 antibody treatment showed prolonged binding of anti-PD-1 antibodies and CD8+ T-cells infiltration [91]. In addition, TREM2-expressing TAMs lead to T-cell exhaustion that can hamper the success of T lymphocyte-based adoptive cell therapies. Depletion of TREM2-expressing TAMs by mAbs was shown to have a robust anti-tumor response alone and in combination with anti-PD1 antibodies through enhanced CD8+ TILs infiltration and effector function in an orthotopic ovarian cancer mouse model [92]. In addition, in a recent study, macrophage depletion using the F4/80 antibody in elderly mice improved the IL-2/anti-CD40 treatment response and achieved up to 78% tumor regression compared to 38% tumor regression in the intact macrophages control and reduced age-related treatment-induced cachexia [93].

### 4.7. Fibrosis and Tissue Remodeling

Chronic wound healing or tissue repair response by fibroblasts after physical, metabolic, immunological, or toxic damage is called tissue fibrosis. Tissue damage by overgrowing tumor cells activates the chronic wound-healing process. This tissue-repairing attempt by the fibroblasts in response to tumor-induced injury is called cancer fibrosis. Cancer-associated fibroblasts (CAF) recruit and activate TAMs by secreting proinflammatory cytokines namely, CCL2, CCL3, CCL5, CXCL8, CSF1, CSF2, IL-6, TGF-β, and VEGF [94,95,96,97]. CAFs are the main regulator of TAM polarization. IL-6 secreted from CAF inhibits the activation of the signal transducer and activator of transcription 3 (STAT3) to block macrophage differentiation [98]. Under the influence of CAF-secreted hypoxia-inducible factor-2α, TAM themselves produce IL-6 to promote macrophage polarization and ARG1 expression [99]. Association between CAF and ECM also impacts TAM behavior. High concentrations of collagen promote TGF-β-mediated M2 polarization of macrophages [100]. Tenascin, hyaluronan, and versican are the common components of tumor-associated fibrosis that regulates macrophage migration and activation. Versican induces the expression of IL-6 and IL-10 by activating TLR2 and activating TAM to promote metastasis [101,102]. Hyaluronan is produced by fibroblast or keratinocytes to recruit TAM and activates them through TLR2 or TLR4 [103,104]. TAM also promotes tissue remodeling by accelerating the angiogenesis and transformation of the adjacent tissue. TAM produces angiogenesis-promoting factors such as EGF, VEGF, PlGF, TGF-β, TNF-α, IL-1β, IL-8, CCL2, CXCL8, and CXCL12 [105,106]. In hypoxic conditions, TAMs tend to produce higher levels of VEGF and promote microvascular density in the TME. WNT7b and HIF-1α expression have a positive correlation with the induced expression of VEGF which plays an important role in the tumor angiogenesis [72,107,108].

### 4.8. Immunosuppression

TAMs play a key role in inhibiting antitumor responses through immunosuppression in a TME. TAM-mediated immunosuppression occurs through the following mechanisms. 

#### 4.8.1. Releasing Anti-Inflammatory Cytokines

Evidence suggests that TAMs, when skewed to pro-tumoral phenotypes (M2-like), modulate the TME, particularly the function of tumor-infiltrating T lymphocytes, favoring pro-tumoral immunoregulation over their anti-tumoral effector functions. TAMs modulate the CD8 T-cell activity by restricting their function by releasing various cytokines. For instance, TAM produces anti-inflammatory and growth-promoting cytokines, such as IL-10, IL-6, TGF-β, and prostaglandin E2 (PGE2), to restrict cytotoxic function of T lymphocytes and promote tumor growth [109]. TAM inhibits the production of IL-12 by limiting the maturation of intra-tumoral DCs by producing a high level of IL-10 which suppress the cytotoxic function of CD8 T-cells [110]. IL-10 also directly restricts the activity of CD8^+^ T-cells by reducing the antigen recognition sensitivity of TCR. IL-10 upregulates the expression of glycosyltransferase, *Mgat5* in CD8^+^ T-cells to induce the N-glycan branching of TCR that restricts the association of co-receptor CD8 with TCR. This event ultimately hampers the TCR signaling, and antigen sensitivity IL-10 also supports the growth of M2 macrophages and suppresses the M1 polarization [111,112,113]. TAM-secreted TGF-β1 downregulates the MiR-34a expression to stimulate the proliferation and invasion of colorectal cancer cells by upregulating the vascular endothelial growth factor [40]. The TGF-β released by TAMs creates an immunosuppressive TME by restricting the CD8^+^ effector T-cell infiltration. M2-type TAMs also recruit and activate Treg cells by secreting CCL22, CCL20, and TGF-β that maintain the inhibitory TME [114]. TAM also downregulates the production of CXCL9 and CXCL10 to restrain CD8 T-cell recruitment [115]. TAM-secreted PGE2 restricts the migration and activation of DC and NK cells in tumors by altering their differentiation and maturation [116,117,118]. A PGE2-mediated delay in DC maturation results in the reduced production of CCL19 which affects the recruitment of naïve T-cells and migration of antigen-carrying DC to lymph nodes [119]. PGE2 induces the Foxp3 expression on T-cells to induce the differentiation of immunosuppressive regulatory T-cells (Tregs); whereas it downregulates the expression of PD-1 in infiltrating CD4 and CD8 cells which results in immune tolerance [120,121]. Moreover, PGE2 alters the production of IL-2 and IFNγ to promote the Th1 to Th2 response [122]. Additionally, PGE2 enhances the migration and activation of myeloid-derived suppressor cells (MDSCs), which plays a central function in sustaining the immunosuppressive microenvironment [123].

#### 4.8.2. Metabolic Reprogramming

In hypoxic conditions, a metabolic shift of tumor cells leads to an accumulation of lactic acid which promotes *Vegf* and *Arg1* expression by TAMs [124]. ARG1 expression is also elevated in TAM after the recognition of apoptotic cell debris by mer tyrosine-protein kinase receptor (MERTKR) [125,126]. Activation of MERTKR in TAM promotes tumor progression by inducing the expression of the proinflammatory cytokines, IL-1β, IL-6, and TNF. An elevated level of ARG1 is a conventional marker of M2-polarized macrophages and is directly related to the suppression of T-cell proliferation [127,128,129]. ARG1 functions in the urea cycle by hydrolyzing L-arginine to L-ornithine and urea. L-arginine is a crucial metabolite for T-cells proliferation and anti-tumor activity, while ornithine supports tumor progression by promoting factors for tissue remodeling and wound healing [130,131,132]. Depletion of L-arginine by induced ARG1 in TAMs leads to the downregulation of CD3ε expression on the T-cells surface which results in TCR inactivation and failure to recognize tumor antigens. L-arginine depletion by ARG1 also inhibits inducible nitric oxide synthase (iNOS) which uses arginine to produce nitric oxide (NO) [130,133,134].

#### 4.8.3. Expressing Immune Checkpoints

Elevated expression of PDL1 in TAM has been reported in multiple tumor tissues, which restrains the activity of tumor-specific effector T-cells in vitro and in vivo conditions by interacting with the PDL1 receptor, PD-1 expressed on T-cells [135,136,137]. Increased expression of TNF-α and IL-10 by monocytes induces the PDL1 expression in TAMs [138]. In addition to PDL1, there are other immune checkpoint ligands expressed by TAMs which directly participate in effector T-cell exhaustion. B7 superfamily member 1 (B7S1, also known as, B7-H4, B7x, or VTCN1) expressed by TAM has a direct role in the dysfunction of tumor-infiltrating T-cells, proinflammatory cytokine production, and cytotoxicity [26,139]. V-domain Ig-containing Suppressor of T-cell Activation (VISTA, also known as, PD-1H, DD1α, c10orf54, Gi24, Dies1, and SISP1) is a homolog of PDL1 which functions as a ligand or inhibitory receptor in myeloid (macrophages) and lymphoid cells (T-cells), respectively. VISTA inhibits the proliferation of antigen-specific T-cells and B-cells, cytokine production, and supporting Treg function [140,141,142,143]. A high VISTA expression is reported in hematopoietic lineage, particularly in tumor-infiltrating myeloid cells, such as myeloid DCs and MDSCs, and Tregs which support VISTA as an important target for immunotherapy. Monoclonal antibody against VISTA reduces the TME immunosuppressive environment by diminishing the TAM and Treg cell infiltration and promoting the effector T-cell function [129,144].

## 5. Therapeutic Targeting of Macrophages

Several macrophage-targeting strategies are currently in the preclinical stages or are being tested in clinical trials. The focus of these strategies has primarily been on depleting tumor-associated macrophages, preventing monocyte recruitment to the tumor, inhibiting macrophage polarization towards the M2 phenotype, or re-educating polarized macrophage so that they can perform anti-tumor functions. Additionally, another focus area of therapeutic targeting using TAMs focuses on blocking inhibitory immune checkpoints, in particular the inhibition of anti-phagocytic immune checkpoints to enhance the phagocytosis of malignant cells. In addition to this, macrophages are also being harnessed as vehicles for delivering cytokines to achieve therapeutic responses and are also being developed as adoptive cell therapies to treat solid tumors (Figure 3). Various approaches to therapeutic targeting and harnessing macrophages have been discussed in this section. A timeline of milestones achieved in macrophage research and their therapeutic targeting is shown in Figure 4.

### 5.1. TAM Depletion

The depletion of TAM by clodronate [152], Zoledronate [15], Trabectedin [153], and FAK inhibitors, such as Defactinib, GSK2256098, and CT-707 (Conteltinib), has been found to inhibit tumor growth in solid cancers [154,155]. Clodronate encapsulated in liposomes effectively depletes macrophages in murine F9 teratocarcinoma and in human A673 rhabdomyosarcoma mouse tumor models and results in the significant inhibition of tumor growth [183]. Moreover, anti-tumor effects of bisphosphonates through TAM depletion have also been demonstrated in a 4T1 mouse breast cancer model [184]. In addition, the targeted TAM depletion or selective depletion of M2 TAMs can be achieved using innovative delivery methods. For example, zoledronate-loaded RBCs used in an innovative TAM-targeted delivery system showed promising results in mouse mammary carcinoma models [185]. Further, lipid-coated calcium zoledronate nanoparticles have shown selective targeting of M2-like TAMs and reduced tumor growth by the inhibition of immunosuppressive effects in a mouse model [15,186]. In another interesting strategy, the selective depletion of M2 TAM was achieved by using bi- and tri-valent T-cell engagers. Activation of endogenous T-cells by CD206- and FRβ-targeting BiTEs/TriTEs exerted preferential killing of M2- over M1-polarized macrophages [187]. Furthermore, TAM depletion by using CAR-T has recently been demonstrated in an orthotopic lung cancer mouse model. Chimeric antigen receptor (CAR) T-cells targeting the macrophage marker F4/80 (F4.CAR-T) effectively killed macrophages in vitro and in vivo without exerting toxicity. F4.CAR-T-cells infiltrated tumor lesions and delayed tumor growth comparably with a PD-1 blockade and significantly extended mouse survival [188]. However, despite these preclinical success stories, the durability of the response to TAM depletion remains below expectations. In addition, TAM depletion has been found to be less effective at preventing cancer growth as compared to the TAM-targeting approaches, such as blocking TAM, recruitment, or reprogramming.

### 5.2. Inhibition of Macrophage Recruitment

Macrophage accumulation in tumors is thought to be a result of the continuous inflow of monocytes from the circulation in response to tumor-derived factors. Colony-stimulating factor-1 (CSF-1) and C−C chemokine ligands, such as CCL2, are crucial tumor-derived factors that mediate the crosstalk between monocytes and tumor cells and foster continuous recruitment and differentiation of monocytes in the TME [189]. Therefore, blocking these factors would be an effective way to prevent the recruitment of TAMs into tumors. Several drugs targeting CSF-1/CSF-1R are currently being investigated in clinical trials (Table 1). PLX3397 (pexidartinib), a CSF-1R inhibitor, has recently been shown to improve the clinical symptoms of patients having Tenosynovial giant cell tumors (TGCT) in a randomized phase III trial [190]. In addition, a phase Ib study has shown a significant reduction in M2-like TAMs at tumor sites in patients with advanced solid tumors when treated with a combination of PLX3397 with paclitaxel [15,191]. In another Phase I/IIa trial, PLX3397 was evaluated in combination with pembrolizumab (PD-1 antibody) for the treatment of advanced melanoma and other solid tumors; however, the study was terminated early due to insufficient evidence of clinical efficacy (NCT02452424). Despite the therapeutic implications of CSF1R inhibitors, clinical trial outcomes based on CSF1R-blocking strategies have proven difficult to improve patient outcomes. Nevertheless, several small molecule and antibody-based CSF-1 inhibitors are also being tested in combination with other therapeutic modalities such as chemotherapy, radiotherapy, and immune checkpoint inhibitors (Table 2).

Likewise, targeting the CCL2-CCR2 axis also reduces the number of M2-like TAMs at primary and metastatic sites, increases CD8 T-cells, and inhibits tumor growth and invasion [192]. Multiple preclinical murine models have demonstrated the potent efficacy of CCR2 inhibitors and anti-CCL2 antibodies in reducing tumor growth and metastasis [193]. The CCL2 inhibitor mNOX-E36 inhibits the recruitment of M2-like TAMs and improves antiangiogenic treatment for glioblastoma in rats [194]. CCX872, another CCL2 antagonist, efficiently reduces tumor-associated MDSCs, which are converted into TAMs in the TME, thereby improving survival in animal models of glioblastoma [195]. A natural CCR2 antagonist (from *Abies georgei* and named 747) showed anti-tumor activity in mouse models of HCC. This was demonstrated by a reduction in the TAM level and concomitant expansion of CD8 T-cells in the TME [196]. In addition, the concurrent administration of anti-CCL2 antibodies was found to improve the efficacy of chemotherapy and checkpoint inhibitors [197] (Table 2). A phase Ib trial using PF-04136309, a CCR2 inhibitor, in combination with chemotherapy FOLFIRINOX (5-fluorouracil, leucovorin, irinotecan, oxaliplatin) in patients with borderline resectable and locally advanced pancreatic adenocarcinoma was found safe and tolerable, however, with a limited response [198]. Moreover, combining the CCR2 antagonist (RS504393) with anti-PD-1 resulted in an enhanced tumor response compared to anti-PD-1 monotherapy in multiple murine tumor models. This combination enhanced anti-tumor responses by increasing CD8+ T-cell recruitment and activation and decreasing CD4+ regulatory T-cells, simultaneously [199]. Moreover, in a preclinical study, a small molecule inhibitor of ASK1(MAP3K5), GS444217, was shown to inhibit ovarian cancer tumor growth by inhibiting the macrophage infiltration in peritoneal ascites by down modulating the activation of endothelial cells [182].

### 5.3. Inhibition of M2 Polarization and Reprogramming

As discussed above, monocytes recruited at the tumor site polarize into M2-like TAMs that promote tumor growth and progression. The reprogramming of M2-like TAMs to convert into M1-like subtypes is another therapeutic approach that is being tested. Puerarin, a MEK/ERK ½ inactivator, has been shown to inhibit M2 polarization in NSCLC [170]. HDAC inhibitors, such as Trichostatin-A [171] and TMP195 [172], promote the M1 polarization of TAMs. Recently, a pharmacological screening of small molecules identified that Doxycycline inhibits the polarization of macrophages towards the M2 phenotype, which in turn limits tumor growth by inhibiting neovascularization [173]. Similarly, STAT-3 inhibitors alone and in combination with the ERK inhibitor reduces TAM polarization [174]. In addition, bisphosphonate zoledronate can also repolarize the TAMs by targeting the mevalonate pathway and inhibiting the development of a mammary tumor [15]. Besides the inhibition of macrophage polarization towards an M2-type, the re-educating or reprogramming of M2 TAMs to behave like M1-type macrophages can also be an effective TAM-targeting strategy that has been focused on in recent years. RP-182, a synthetic peptide, reprograms M2 macrophages by activating the mannose receptor CD206 expressed on the M2 macrophage and turning them into M1-like phenotypes, thereby limiting tumor progression [175]. *Duvelisib* (IPI-145), an oral inhibitor of the PI3Kδ and PI3Kδγ isoforms, can induce the M2- to M1-like reprogramming of macrophages [176,177]. CD40 agonistic mAbs can induce TAMs to secrete high levels of matrix metalloproteinase 13 (MMP13), an enzyme that supports tumor control by degrading fibrotic tissue [178]. Multiple agonists of TLR7,8,9 have been shown to skew anti-tumor phenotypes in TAMs [179]. Clinical trials are also underway for TLR agonists to treat solid tumors (Table 1). TLR9 agonists, such as lefitolimod, effectively modulate the TME and induce anti-tumor responses by promoting the infiltration of CD8 T-cells and reprogramming TAMs in the TME. *Ibrutinib,* a Btk inhibitor has been shown to reprogram TAMs in pancreatic cancer [180]. In addition, the nanoparticles-based delivery of TLR agonists, bisphosphonates, DNA, mRNA, and miRNA repolarizes TAMs more effectively. A comprehensive review has recently been published on this topic [181].

### 5.4. Targeting Programmed Cell Removal (PrCR) and Anti-Phagocytic Checkpoints

Macrophages remove damaged, dysfunctional, aging, or harmful cells by phagocytosis called PrCR. The PrCR process involves recognizing, engulfing, and digesting target cells intracellularly. Several pro-phagocytic signals called “Eat me signals” have been identified that facilitate the PrCR of cancer cells. However, by expressing “Don’t eat me” signals, cancer cells counterbalance this elimination mechanism and evade immune clearance. An “Eat me signal” calreticulin secreted or exposed on the cell surface of cancer cells allows macrophages to recognize and phagocytose them [89,200]. However, this effect is antagonized by CD47, a “don’t eat me” signal often overexpressed by cancer cells. Through the interaction with its receptor, SIRPα, CD47 activates anti-inflammatory signaling in macrophages, causing them to cease phagocytosis. Further, PrCR is also induced by several chemotherapeutic agents, including docetaxel, doxorubicin, carboplatin, and cisplatin. This phenomenon is called immunological cell death. The dying tumor cells release tumor antigens and adjuvant molecules (ATP, HMGB1, and ecto-calreticulin) that trigger the involvement of macrophages in the immune response [164,165,166]. Docetaxel is a chemotherapeutic drug known for its antimitotic activity; however, recent studies have highlighted its immunomodulatory role. Docetaxel induces proinflammatory chemokine (C–C motif) ligand 3 (CCL3), which induces proinflammatory macrophage differentiation and phagocytosis [201]. In addition to this, docetaxel also induces calreticulin translocation to the plasma membrane of the dying cells and triggers phagocytosis of the cancer cells by TAMs. In a recent study, docetaxel and its combination with carboplatin or cisplatin significantly increased ATP levels, ecto-calreticulin expression, and HMGB1 expression in NSCLC cell lines, leading to the phagocytosis of treated cells and maturation of DCs [202]. Moreover, at the molecular level, TLR pathways and the subsequent activation of Bruton’s tyrosine kinase (Btk) signaling induces PrCR in macrophages by regulating the phosphorylation and surface trafficking of calreticulin [203]. Various therapeutic approaches to harness the ability of programmed cell removal by macrophages are in various stages of clinical development. A large number of inhibitors including monoclonal antibodies and soluble peptides directly targeting CD47 and SIRPα are in clinical trials. (Table 1). In addition, several other anti-phagocytic surface proteins (immune checkpoint) such as PDL1, LILRB1, B2M, and CD24 have also been found to inhibit the phagocytosis of cancer cells both in the in vitro co-culture of cancer cells with macrophages as well as in vivo. The inhibition of this immune checkpoint using monoclonal antibodies has shown promising results in preclinical studies. Some inhibitors such as (Anti-LILRB2) are also in clinical trials (Table 1).

### 5.5. Macrophages-Mediated Cytokine Delivery

A recent study by De Palma et al., used macrophages to deliver IFNγ to tumor sites. They transferred the *Ifna1* gene into hematopoietic progenitors under the promoter of the *Tie2* gene. Due to the high migratory and tumor-homing abilities of Tie2-expressing monocytes, the cell-specific expression of IFNγ in Tie2+ monocytes allowed the targeted release of IFNγ at the tumor sites [168]. The TME delivery of IFNγ inhibited tumor growth and angiogenesis by triggering the immune response. TEMFERON, genetically modified Tie2-expressing monocytes (TEMs), targeting interferon α-2 (IFNα-2) expression in GBM TME is being tested in a clinical trial (Table 1). Similarly, in another approach, IFNα containing soft particles called *backpacks* coated on the macrophage surface significantly reduced tumor growth and metastatic burden when injected intra-tumorally [169].

### 5.6. Macrophages-Based Adoptive Cell Transfusion

T lymphocyte-based adoptive cell therapies such as CAR-T-cells or TCR-engineered T-cells have shown remarkable anti-tumor responses in different advanced hematological malignancies, as evidenced by six FDA approvals in the last 10 years [204]. These therapeutic modalities are, however, not yet available in clinical trials for solid cancers. The low response of adoptive cell therapies in solid tumors is because of the limited penetration of adoptively transferred cells into the tumor sites. The barrier to this immune infiltration is usually immunosuppression and fibrotic tissue remodeling. As discussed above, TAMs are the major cause of tumor immune suppression and fibrosis; therefore, their depletion and inhibition of recruitment is being extensively investigated. However, unlike lymphocyte-based cellular therapeutic agents, macrophages have superior trafficking and homing capabilities in solid tumors which makes them a powerful therapeutic tool. Recently, Michael Klichinsky and colleagues pioneered in developing a CAR-M platform to treat HER2 positive solid tumors. The expression of CAR19ζ and HER2 ζ CARs in THP-1 cells specifically killed CD19^+^ K562 and HER2^+^ SKOV3, respectively, in an in vitro co-culture through enhanced phagocytosis. Further, a significantly reduced tumor growth and increases overall survival were demonstrated in two mice xenograft models and a humanized mouse model [151]. Currently, Carisma Therapeutics Inc. is conducting the first ever CAR-M clinical trial in humans for HER2 positive solid tumors, and it is presently in phase I (NCT04660929, recruiting).

## 6. Challenges in Therapeutic Targeting of TAMs

Macrophages are one of the most plastic cell types of the immune system. The local microenvironment in which they reside determines their polymorphism and functional heterogeneity. Macrophages constitute the most abundant population of immune origin in solid tumors. In a variety of human cancers, TAMs seem to be associated with a poor prognosis, although it varies by cancer type or context. The role played by TAMs in tumor progression makes them an attractive target for anti-tumor treatments [4,12,17,131]. There have been several therapeutic strategies developed that directly target TAMs or their functional mediators. These strategies include the depletion of TAMs, the blocking of monocyte recruitment, the reprogramming of TAMs into proinflammatory M1 macrophages, and the neutralization of their products [17,39,205]. Several antagonists that target TAMs have already been tested in various clinical trials, even though most TAM-targeting strategies are still in the preclinical stages [8,15,89,92,186,206,207,208,209,210]. There are still a lot of obstacles to overcome and many issues to be resolved before targeting TAMs becomes a reality. Finding the real prognostic value of TAMs in various cancers is a big challenge. There are disagreements among different studies regarding the prognostic value of TAMs in solid tumors. This is because different studies find different prognostic values of TAMs, not only depending on the cancer type, but also for the same type of cancer. For example, in the case of a highly heterogenous cancer such as gastric cancer, bladder cancer, and pancreatic cancer, both positive and negative prognostic values of TAMs have been reported [9,10,23,24,31,209]. These studies were limited by the poor understating of TAM heterogeneity and the use of M2 markers as the sole predictor. Hence, a further sub-typing of TAMs beyond an M1/M2 dichotomy can present a clearer picture of TAM density-based disease prognosis. Further, identifying the appropriate stage for TAM-targeted interventions to achieve the maximum therapeutic response can be achieved by better understanding the roles of TAMs in cancer progression from early-stage initiation to later metastatic stages. For example, a transition to M2 macrophages occurs in the advanced stages of cancer, whereas M1 macrophages dominate in the early stages [199]. Accordingly, TAM repolarization or M2 TAM depletion may be more effective during the advanced stages, whereas TAM activation may be more effective during the early stages when M1 macrophages dominate.

## 7. Conclusions

The macrophage is the most abundant and universal cell type in TMEs of solid tumors. In certain cancer types, TAMs can even outnumber cancer cells in the later stages of the disease. Macrophages are engaged in a complex interaction with cancer cells, stromal cells, and immune effector cells in TMEs. Despite being part of the immune system, their inability to prevent cancer growth and, rather, fueling it, merit attention. In the past few years, several mechanisms have been uncovered that suggest that TAMs may have a vital role in stimulating tumor growth. A negative correlation between TAM density and disease outcome has been found in most cancer types, except a few that make them an attractive therapeutic target. The principle of targeting cancer by interfering with TAMs and their effector molecules has certainly shown some promise in recent years. However, success is hampered and limited by the need for a detailed phenotypic characterization of macrophages in a specific cancer type. However, using checkpoint blockade-based immunotherapies and other treatment modalities in conjunction with TAM-based interventions presents hope for improved anti-cancer treatments, particularly for solid tumors. Further, despite the remarkable success of adoptive cell therapies such as CAR-T-cell therapy in the treatment of hematological malignancies, there has been very limited success in the treatment of solid tumors due to the poor recruitment of T-lymphocytes at tumor sites or lympho-exclusion by immunosuppressive TMEs in solid tumors. Since TAMs are continuously replenished by circulating myeloid precursor cells, the use of macrophages as cell therapies may be able to overcome these limitations. The development of the first CAR-M therapy has laid the foundation for further advancements in macrophage-based future cancer immunotherapies. Furthermore, macrophages have been identified as a vehicle to deliver cytokines to tumor sites. Nonetheless, further characterization of TAM marker genes and pro-tumorigenic TAM-specific promoters will be helpful in formulating future macrophage-based immunotherapies.

## Figures and Tables

**Figure 1 vaccines-11-00055-f001:**
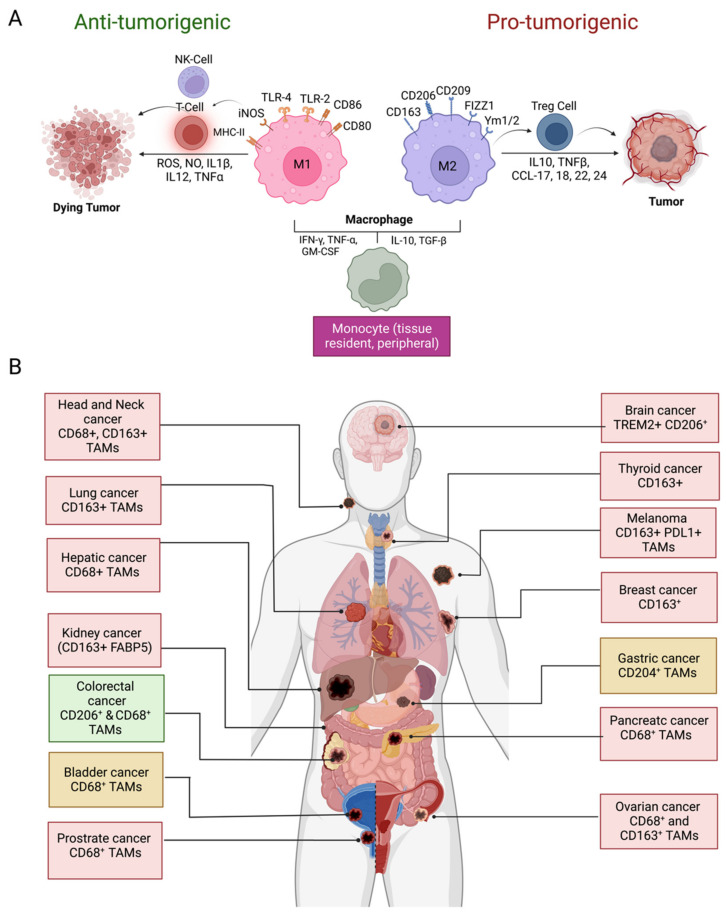
Macrophage polarization and prognostic significance in cancer. (**A**) Differentiation of monocytes in M1 and M2 macrophages in the TME and their role in tumor suppression and promotion, respectively; (**B**) prognostic significance of TAM density in various cancer types and IHC markers used for TAM identification. Presence of TAMs correlates with the unfavorable disease prognosis in glioblastoma, thyroid, lung, hepatic, kidney, ovarian, head and neck, breast, prostate, and melanoma (red boxes) whereas favorable prognosis (green box) in colorectal cancer. Both favorable and unfavorable correlations have been reported in gastric cancer and bladder cancer (yellow boxes). Abbreviations: NK, natural killer; ROS, reactive oxygen species; NO, nitric oxide; IL, interleukin; TNF, tumor necrosis factor; MHC, major histocompatibility complex; iNOS; inducible nitric oxide synthase; TLR, toll-like receptor, IFN, interferon; TGF, tumor growth factor; GM-CSF, colony-stimulating factor; Treg, regulatory T lymphocytes; FIZZ1, resistin-like molecule α, Ym1/2; chitinase-like protein Ym1/2, CCL; C-C chemokine ligand; CD, cluster of differentiation; FABP5, fatty acid binding protein 5; TREM, triggering receptor expressed on myeloid cells; PDL1, programmed death ligand 1 (Figure created with BioRender.com accessed on 15 December 2022).

**Figure 2 vaccines-11-00055-f002:**
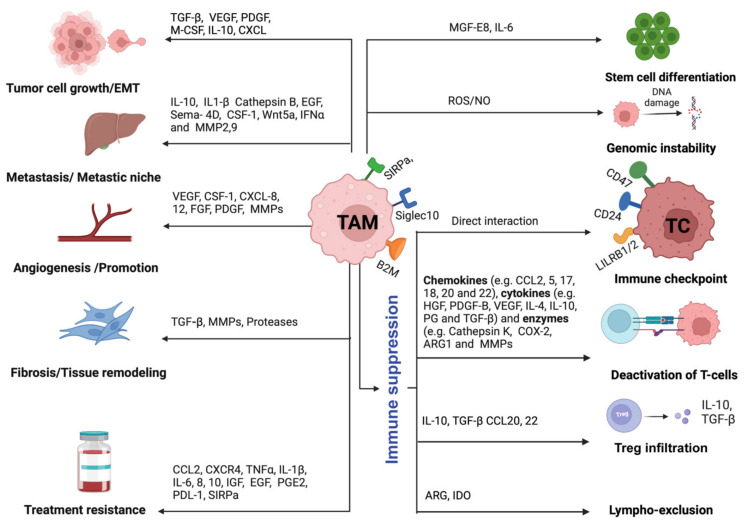
Illustration showing the role of tumor-associated macrophages in tumor progression through various mechanisms such as tumor growth/EMT, metastatic niche formation, initiation and promotion of angiogenesis, tumor tissue remodeling and fibrosis, treatment resistance, stem cell differentiation, genomic instability, and immune suppression. Abbreviations: ARG1, Arginine 1; TAM, tumor-associated macrophages; TC, tumor cell; B2M, beta-2-microglobulin; CCL, C-C chemokine ligand; COX-2, cyclooxygenase-2; CSF-1, colony-stimulating factor-1; CXCL, CXC chemokine ligand; CXCR, CXC chemokine receptor; EGF, epidermal growth factor; FGF, fibroblast growth factor; HGF, hepatocyte growth factor; IDO, indoleamine 2,3-dioxygenase; IFN, interferon; IGF, insulin-like growth factor; IL, interleukin; LILRB1/2, leukocyte immunoglobulin-like receptor subfamily B member 1/2; MFGE8, milk fat globule EGF and factor V/VIII domain containing; MMP, matrix metalloproteinase; M-CSF, Macrophage colony-stimulating factor; NO, nitric oxide; PDGF, platelet-derived growth factor; PDL1, programmed death ligand 1; PGE2, prostaglandin E2; ROS, reactive oxygen species; SEMA4D, semaphorin 4D; SIRPα, signal regulatory protein alpha; TGF-β, transforming growth factor-β; TNF, tumor necrosis factor; VEGF, vascular endothelial growth factor (Figure created with BioRender.com accessed on 15 December 2022).

**Figure 3 vaccines-11-00055-f003:**
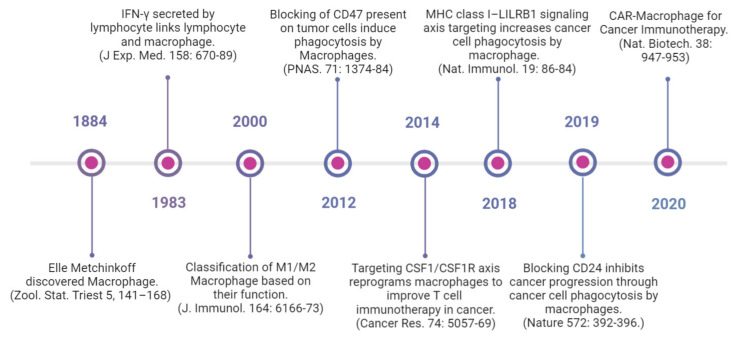
Timeline showing advancements in macrophage research and their therapeutic targeting (Created with BioRender.com accessed on 15 December 2022) [14,145,146,147,148,149,150,151].

**Figure 4 vaccines-11-00055-f004:**
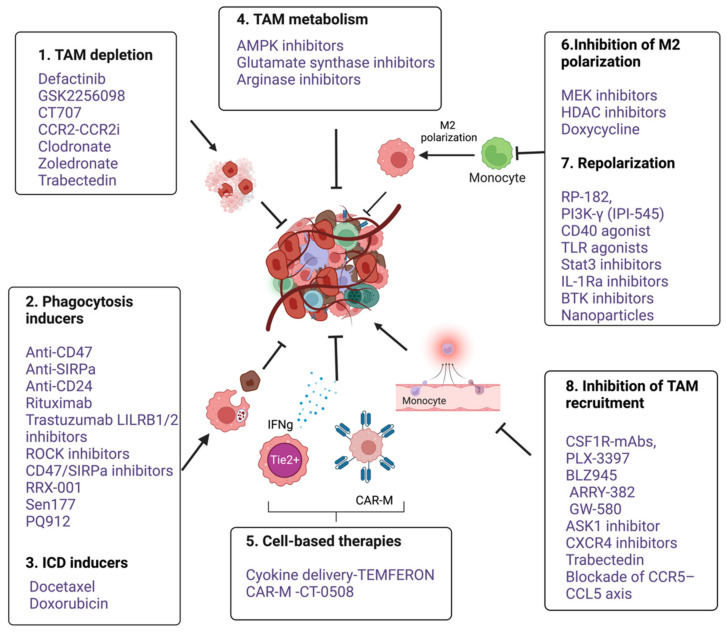
An overview of various anti-cancer immunotherapeutic modalities targeting tumor associated macrophages. 1. TAM depletion: Bisphosphonates such as clodronate-liposome [152], Zoledronate [15] and cytotoxic drug Trabectedin [153] and FAK inhibitors such as Defactinib, GSK2256098 and CT-707 (Conteltinib) [154,155], 2. Phagocytic inducers; Monoclonal antibodies targeting myeloid checkpoints such as CD47 and SIRPα in lymphomas, Anti-CD24 in ovarian and breast cancer [14], Rituximab targeting CD20 in B cell lymphoma [156,157], Trastuzumab targeting HER2 in breast and ovarian cancers [158,159]. LILRB1/2 inhibitors [160]. Small molecule inhibitors targeting CD47- SIRPα axis such as RRX-001 [161], Sen177 and PQ912 [162], Rho-kinase inhibitor (Y27632) in solid cancer models [163]. 3. ICD inducers; docetaxel, doxorubicin [164,165,166]. 4. Targeting TAM metabolism; Metformin, an AMPK inhibitor, 2-Deoxy-d-glucose (2DG), a Hexokinase-2 inhibitor [167]; Glutamine Synthetase inhibitor, Methionine Sulfoximine and arginase inhibitors (CB-1158 and L-Norvaline). 5. Cell based therapies- Macrophages as delivery vehicle; TEMFERON, TME delivery of IFNγ [168], and IFNα [169], CAR macrophages; CT-0508, targeting solid tumors [151]; 6. Inhibition of M2 polarization. Puerarin [170]. Trichostatin-A [171] and TMP195 [172], Doxycycline [173]. Stat-3 inhibitors alone and in combination with ERK inhibitor [174]. 7. Repolarization from M2 to M1; RP-182, a synthetic peptide [175]. Duvelisib (IPI-145), an oral inhibitor of the PI3Kδ and PI3Kδγ isoforms [176,177]. CD40 agonistic mAbs [178]. TLR7,8,9 agonist [179]. Ibrutinib, a BTK inhibitor [180]. Nanoparticles based delivery of TLR agonists, Bisphosphonates, DNA, mRNA, and miRNA [181]. 8. Inhibition of TAM recruitment; Antagonists of CSF1R, CXCR4, CCR5-CCL5 and CCL2-CCR2 (Table 1). GS444217, an ASK1(MAP3K5) inhibitor [182] (Figure created with BioRender.com accessed on 15 December 2022).

**Table 1 vaccines-11-00055-t001:** Selected clinical trials of TAM-targeting drugs.

Targeting TAMs Strategies	Name	Targets	Cancer Types	Phases	Clin. Trial
Phagocytosis	RRX-001	CD47, SIRPα	Non-small cell lung cancer	Phase III	NCT03699956
	Hu5F9-G4	CD47	Advanced tumors	Phase I	NCT02216409
	JTX8064	LILRB2	Advanced refractory solid tumors	Phase I	NCT04669899
Depletion of M2-like TAMs	Zoledronate	NA	Mammary carcinoma	Phase III	NCT00320710
TAMs recruitment	Pexidatintinib	CSF-1R	Advanced solid tumors	Phase III	NCT02371369
	D2923	CSF-1R	Myelogenous leukemia	Phase II	NCT04989283
	Emactuzumab	CSF-1R	Advanced solid tumors	Phase III	NCT05417789
	3D185	CSF-1R	Colorectal cancer	Phase II	NCT05039892
	PLX3397 (Pexidarnitib)	CSF-1R	Melanoma	Phase II	NCT02071940
		CSF-1R	Advanced solid tumors	Phase I	NCT02734433
		CSF-1R	PVNS or GCT-TS	Phase III	NCT02371369
		CSF-1R	Leukemia, sarcoma, or neurofibroma	Phase I/II	NCT02390752
		CSF-1R	Acute myeloid leukemia	Phase I/II	NCT01349049
	PLX7486 (Plexxikon)	CSF-1R		Phase I	NCT01804530
	DCC-3014	CSF-1R CSF-1R	Advanced-stage or metastatic solid tumors	Phase I	NCT03069469
	ARRY-382			Phase I	NCT01316822
	LY3022855 mAb (IMC-CS4)	CSF-1R	Solid tumors	Phase I	NCT02265536
				Phase I	NCT01346358
	AMG820 mAb	CSF-1R	Solid tumors	Phase I/II	NCT01444404
	MLN1202	CSF-1R	Bone metastasis	Phase I/II	NCT01015560
	AMG820 mAb/Pembrolizumab	CSF-1R	Solid tumors	Phase I/II	NCT02713529
	BLZ945/PRD001	CSF-1R	Advanced solid tumors	Phase I/II	NCT02829723
	Cabiralizumab/Nivolumab	CSF-1R	Advanced solid tumors	Phase I	NCT02526017
	MLN1202	CSF-1R	Bone metastasis	Phase I/II	NCT01015560
	RO5509554/RG7155 (Emactuzumab)/Paclitaxel	CSF-1R	Advanced solid tumors	Phase I	NCT01494688
	PD-0360324 mAb/Cyclophosphamide	CSF-1R	Ovarian cancer	Phase II	NCT02948101
	Eribulin	CSF-1R	Metastatic breast cancer	Phase I/II	NCT01596751
	PF-04136309	CCR2	Pancreatic cancer	Phase II	NCT02732938
	CCX872	CCR2	Pancreatic cancer	Phase I	NCT02345408
	CCR2i	CCR2	Cutaneous t-cell lymphoma	Phase II	NCT02732938
	mNOX-E36	CCL2	Glioblastoma	Phase I	NCT00976729
	Carlumab (anti-CCL2 antibodies Centocor)	CCL2	Prostate cancer	Phase II	NCT00992186
	CNTO 888 (Carlumab)	CCR2	Prostate cancer	Phase II	NCT00992186
	PF-04136309	CCR2	Pancreatic cancer	Phase I/II	NCT02732938
TAMs reprogramming	R848	TLR7/8	Colorectal cancer	Phase II	NCT00960752
	lefitolimod	TLR9	Small-cell lung cancer	Phase I	NCT02668770
	RP6530	PI3Kδ/γ	Hodgkin lymphoma	Phase I/II	NCT03770000
Cell-based therapies	CAR-M	HER2	Solid cancers	Phase I	NCT04660929
	TEMFERON	Tie-2	Glioblastoma multiforme	(Phase I/IIa)	NCT03866109

**Table 2 vaccines-11-00055-t002:** Selected clinical trials of the TAM-targeting agents in combination with other therapeutic interventions.

CSF-1R Inhibitors + Checkpoint Immunotherapy				
Drug Name	**Combination Drugs**	Cancer Types	Phasess	Clin. Trials
PLX3397 (Pexidarnitib)	Pembrolizumab	Solid tumors	Phase I/II	NCT02452424
	Durvalumab	Advanced tumors	Phase I	NCT02777710
LY3022855 mAb (IMC-CS4)	Pembrolizumab	Pancreatic cancer	Phase I	NCT03153410
	Durvalumab			
	Tremelimumab	Advanced solid tumors	Phase I	NCT02718911
RO5509554/RG7155 (Emactuzumab)	Atezolizumab	Solid tumors	Phase I	NCT02323191
CSF-1R Inhibitors + Chemotherapy				
PLX3397 (Pexidarnitib)	Paclitaxel	Advanced solid tumors	Phase I/II	NCT01525602
	Standard Chemotherapy			NCT01042379
CSF-1R Inhibitors + Targeted Therapy				
PLX3397 (Pexidarnitib)	Sirolimus (Rapamycin)	Sarcoma		NCT02584647
CSF-1R Inhibitors + Radiotherapy				
PLX3397 (Pexidarnitib)	RT + ADT	Prostate cancer	Phase I	NCT02472275
	RT + Temozolomide	Glioblastoma	Phase I/II	NCT01790503
CCR2/CCR5 Inhibitors + Checkpoint Immunotherapy				
BMS-813160 (CCR2/CCR5 antagonist)	Nivolumab/Nabpaclitaxel	Advanced solid tumors	Phase I/II	NCT03184870
	Nivolumab/iplimumab		Phase II	NCT0299611
	Nivolumab	Hepatocellular carcinoma	Phase II	NCT04123379
CCR5 antagonist	Pembrolizumab	CRC	Phase I	NCT03274804
	Nivolumab plus Ipilimumab	Pancreatic cancer, CRC	Phase I	NCT04721301
CCR2 Inhibitors + Chemotherapy				
CNTO 888 (Carlumab)	Gemcitabine/paclitaxel	Advanced solid tumors	Phase II	NCT01204996
	Carboplatin/doxorubicin			
PF-04136309	FOLFIRINOX	Advanced solid tumors	Phase I/II	NCT01413022
Anti-CD47/SIRPα antibodies+ Other immunotherapies				
Hu5F9-G4	Pembrolizumab	Solid tumors	Phase I/II	NCT03869190
	Multiple immunotherapy	Urothelial and bladder cancer	Phase I/II	NCT03869190
BI 754,091 (OSE Immunotherapeutics)	BI 754,091 (anti-PD1)	Solid tumors	Phase I	NCT03990233

## Data Availability

Not applicable.

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
