# Peer review of "Macrophages as a Potential Immunotherapeutic Target in Solid Cancers"

_vaccines, 2022, doi:10.3390/vaccines11010055_

Round 1

Reviewer 1 Report

The manuscript entitled: “Macrophages as a Potential Immunotherapeutic Target in Solid Cancers” by Alok K. Mishra et al. reviews the pathobiology of TAMs and significance of potential therapeutic strategies for targeting TAMs in cancer patients.

Albeit the review is well written, prepared and of special interest, some comments should be addressed.

Comments:

1.    Page 16, point 6: Please highlight in more detail the potential obstacles for targeting TAMs and which next steps are needed.

2.    Conclusions: the conclusions should focus in more detail on the unmet clinical need in oncology. What can be learnt from this review, what are potential further/future investigations

3.    Figure 3 and 4. Please choose a higher resolution for the figures.

4.    Table 1. Please add references of each study directly in the table.

Author Response

Reviewer 1

The manuscript entitled: “Macrophages as a Potential Immunotherapeutic Target in Solid Cancers” by Alok K. Mishra et al. reviews the pathobiology of TAMs and significance of potential therapeutic strategies for targeting TAMs in cancer patients. Albeit the review is well written, prepared and of special interest, some comments should be addressed.

Comments:

  1. Page 16, point 6: Please highlight in more detail the potential obstacles for targeting TAMs and which next steps are needed.

Authors response: The suggested changes have been made in the revised manuscript

  1. Conclusions: the conclusions should focus in more detail on the unmet clinical need in oncology. What can be learnt from this review, what are potential further/future investigations

Authors' response: The suggested changes have been made in the revised manuscript

  1. Figure 3 and 4. Please choose a higher resolution for the figures.

Authors' response: high-resolution files of the figure have been separately attached

  1. Table 1. Please add references of each study directly in the table.

Authors' response: We apologize for not clearly understanding the intent of the reviewer in this comment. The NCT number of each clinical trial has been given in the table

Reviewer 2 Report

1. In line 38 of page 1, it would be more appropriate to replace Toll like receptors (TLRs) with Pattern recognition receptors (PRRs)

2. In “pro-tumorigenic roles of TAMs” section, some cytokines and chemokines appear repeatedly, and I suggest that the authors could make this section more condensed.

3. Abbreviation that appear for the first time should be clearly write with the full name, but the abbreviations and their full name appear together for several times in the manuscript, please check throughout the manuscript and revise it.

4. There are some clerical errors in the manuscript.

Author Response

Reviewer 2

  1. In line 38 of page 1, it would be more appropriate to replace Toll like receptors (TLRs) with Pattern recognition receptors (PRRs)

       Authors response : Thanks for the valuable suggestion. We have made this change to the revised manuscript.

  1. In “pro-tumorigenic roles of TAMs” section, some cytokines and chemokines appear repeatedly, and I suggest that the authors could make this section more condensed.

Authors' response: Thanks for the valuable suggestion. We have thoroughly revised the manuscript in order to remove any redundant information.

  1. Abbreviation that appear for the first time should be clearly write with the full name, but the abbreviations and their full name appear together for several times in the manuscript, please check throughout the manuscript and revise it.

Authors' response: We have carefully revised the inconsistencies with abbreviations

  1. There are some clerical errors in the manuscript.

Authors' response: We have carefully revised and made our best efforts in revising the manuscript to avoid any errors.